# PERSONBIAS: A LIGHTWEIGHT FRAMEWORK FOR PERSONALIZED BIAS MITIGATION IN LARGE LANGUAGE MODELS

## ABSTRACT

Social bias in large language models (LLMs) outputs has emerged as a critical challenge in artificial intelligence. While existing bias detection methods pursue comprehensive identification and elimination of implicit biases, this *one-size-fits-all* approach presents significant limitations. Excessive bias correction causes responses to deviate from user query intent, comprehensive detection demands extensive human annotation and computational resources, and critically, user heterogeneity dictates that different individuals with diverse backgrounds and personality traits exhibit varying sensitivities toward different bias types. To address these challenges, we propose PersonBias, a lightweight, personalized debiasing framework that balances bias mitigation with response quality optimization. Our approach leverages LLMs to automatically extract user personality features from conversational contexts, eliminating the need for explicit demographic data collection. We develop a dual-tower encoder architecture with cross-attention mechanisms to model user-specific bias sensitivities, employing parameter-efficient fine-tuning that freezes encoder parameters while optimizing only projection layers and attention mechanisms. Rather than requiring model-specific fine-tuning, PersonBias operates through real-time intervention during generation, dynamically evaluating and adjusting outputs at fixed token intervals to prevent bias accumulation while maintaining relevance and utility. Experiments on multi-turn dialogue datasets demonstrate that PersonBias achieves superior bias reduction and utility preservation compared to prompt-based and fine-tuning baselines, offering a practical and adaptive solution for personalized fairness in LLMs.

## 1 INTRODUCTION

In recent years, Large Language Models (LLMs) (Achiam et al., 2023; Liu et al., 2024a; Bai et al., 2023) have demonstrated remarkable reasoning and emergent capabilities, leading to their widespread adoption across diverse domains. Despite their impressive performance, recent studies have revealed that LLMs exhibit systematic social biases against certain demographic groups during response generation. This phenomenon significantly impedes the deployment and adoption of LLMs across different geographical regions and application domains. (Lin & Li, 2025; Gallegos et al., 2024a) Consequently, effectively mitigating multiple biases in LLM generation processes has emerged as a critical challenge in fairness research for large language models.

Current debiasing approaches for LLMs can be categorized into two primary paradigms: fine-tuning-based methods and prompt-based methods. Fine-tuning-based approaches typically employ reinforcement learning strategies such as Reinforcement Learning from Human Feedback (RLHF) (Casper et al., 2023) to model biases, followed by Parameter-Efficient Fine-Tuning (PEFT) (Han et al., 2024) strategies to fundamentally mitigate model biases (Tan et al., 2024b; Wagner et al., 2025). While these methods demonstrate substantial effectiveness, they require extensive human-annotated high-quality bias datasets and substantial computational resources. Consequently, prompt-based methods (Zhang et al., 2025a; Furniturewala et al., 2024) are often adopted in resource-constrained environments or when dealing with black-box models. These approaches typically leverage carefully crafted prompts or model-generated prompt templates to guide LLMs, thereby intervening in the generation process to produce unbiased responses. While existing methods have

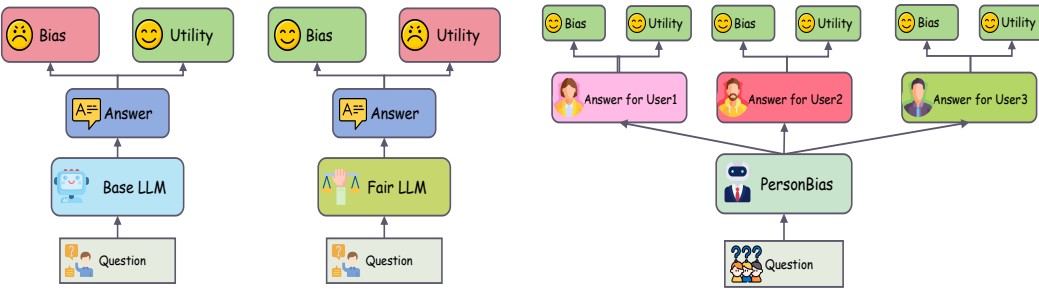

Figure 1: Problems of *one-size-fits-all* Debiasing.   Figure 2: Core Idea of PersonBias.

achieved notable progress in eliminating language model biases, these *one-size-fits-all* debiasing strategies exhibit significant limitations. First, as shown in Fig.1, from the perspective of user requirement adaptation, models that pursue absolute fairness often impose complex constraints on LLM outputs, potentially causing responses to deviate from users' actual needs (Lin et al., 2025). Moreover, the inherent heterogeneity of user populations indicates substantial variations in sensitivity and tolerance thresholds for different types of biases across users with diverse backgrounds and individual characteristics. Second, regarding implementation feasibility, fine-tuning-based debiasing methods typically demand large-scale annotated data and prohibitive computational costs, making them impractical in resource-constrained environments or when dealing with black-box models. Third, considering temporal dynamics, biases in multi-turn dialogues or long-text generation scenarios often manifest through progressive revelation and cumulative amplification (Li et al., 2025; Cheng et al., 2025). Existing prompt-engineering methods lack real-time monitoring capabilities for the generation process, potentially leading to gradual bias amplification. Based on this analysis, LLM debiasing research faces three core challenges: (1) how to construct personalized debiasing frameworks that accommodate heterogeneous user requirements, (2) how to implement lightweight yet effective debiasing mechanisms under resource constraints, and (3) how to establish bias detection and correction systems that integrate real-time monitoring with adaptive adjustment.

To address these challenges, we propose a Personalized Debiasing framework named PersonBias, whose core architecture comprises three key components, Fig.2 illustrates the core idea of PersonBias. First, we construct a user profile extraction module that leverages LLMs to extract personalized characteristics from historical dialogue data. Second, we design a personalized preference reward model employing a dual-tower encoder architecture with independent encoders for user information and textual content. Multi-head cross-attention mechanisms capture correlation patterns between user features and textual bias information to generate personalized evaluation scores. We adopt parameter-efficient fine-tuning, freezing encoder parameters while optimizing only mapping layers, prediction layers, and attention mechanisms. Finally, we introduce a dynamic debiasing strategy that periodically evaluates real-time generated content using the personalized reward model, filtering low-scoring text while retaining high-quality outputs to ensure optimal user satisfaction. Our main contributions are:

- We propose a personalized debiasing framework (PersonBias) that automatically extracts user characteristics from conversational history and employs a lightweight dual-tower encoder architecture with cross-attention mechanisms to model user-specific bias sensitivities, enabling tailored bias detection without requiring explicit demographic data collection.

- We introduce a dynamic bias monitoring strategy that performs real-time intervention during text generation by periodically evaluating candidate responses at fixed token intervals, preventing bias accumulation while preserving response relevance and utility.

- Experiments on multi-turn dialogues show that PersonBias significantly mitigates bias while preserving utility, outperforming both prompt-based and fine-tuning methods to provide a practical, adaptive path toward fairness in LLMs.

## 2 RELATED WORK

### 2.1 BIAS MITIGATION IN LANGUAGE MODELS

Eliminating bias in language model responses represents a critical challenge in current research on model fairness. Recent debiasing research for language models primarily employs two methodological approaches: fine-tuning-based and prompt-based debiasing techniques. **Fine-tuning-based methods** adjust model parameters through various specialized tuning strategies and architectural designs, including reinforcement learning-based fine-tuning (Fan et al., 2025; Tomar et al., 2025), causal theory-informed fine-tuning(Sun et al., 2024; Wu et al., 2024), continual debiasing strategy (Lee et al., 2025; Kim et al., 2024) and module-level intervention strategies(Cheng et al., 2025; Chen et al., 2023). Additionally, some approaches utilize LLMs or manually curated comprehensive bias datasets (Fan et al., 2024b; Zhou et al., 2025; Fan et al., 2024a) to either fine-tune models or serve as external knowledge bases for bias detection guidance. In contrast, **Prompt-based methods** (Yang et al., 2025; Zhang et al., 2025a; Chisca et al., 2024; Yang et al., 2023) leverage carefully engineered prompts and debiasing data as guidance mechanisms (Furniturewala et al., 2024; Gallegos et al., 2024b), enabling language models to generate equitable responses while circumventing complex fine-tuning procedures. These methods fail to account for the potential response deviation or even irrelevance caused by a *one-size-fits-all* debiasing strategy during the debiasing process.

### 2.2 PERSONALIZED LLMS

Due to user heterogeneity and varying requirements for large language models, personalized LLMs have become a prominent research direction (Liu et al., 2025; Zhang et al., 2025b). Existing approaches can be categorized into two main classes: (1) **Fine-tuning-based Personalized LLMs**: The core strategy of this approach involves training dedicated LLM models tailored to individual users based on their personalized data (Tan et al., 2024b;a). Given the substantial computational costs associated with full fine-tuning, current research in fine-tuning-based personalized LLMs predominantly focuses on exploring Parameter-Efficient Fine-Tuning (PEFT) methods (Wagner et al., 2025; Peng et al., 2024) to achieve cost-effective personalization of large language models. (2) **Retrieval-based Personalized LLMs**: Considering the prohibitive costs of fine-tuning and the practical deployment challenges of maintaining dedicated models for all users, retrieval-based personalized LLM approaches have emerged, inspired by Retrieval-Augmented Generation (RAG) techniques (Lewis et al., 2020). The primary mechanism of these methods involves retrieving relevant documents from users' historical interaction records that correspond to current queries, and constructing personalized contextual prompts for LLMs based on these retrieved documents to enable personalized response generation (Salemi et al., 2024; Tang et al., 2024; Zhu et al., 2025). It is noteworthy that existing personalized approaches primarily concentrate on user recommendation systems, with relatively limited attention devoted to the debiasing domain, thereby presenting significant opportunities for future research endeavors.

## 3 PROBLEM FORMALIZATION

The core objective of debiasing strategies for LLMs is to comprehensively identify and eliminate biases present in model responses. Such methods typically train a reward model based on a bias dataset or directly employ the dataset as an external knowledge base to accurately detect latent bias information in responses of the base model $\pi_{base}$ to input $X$, subsequently guiding the model to regenerate answers while avoiding similar biased terms. This process can be formalized as follows:

$$S = \mathcal{D}\left(\pi_{base}\left(X\right), B\right), \tag{1}$$

where $S$ is the bias score of a response, quantifying the degree of bias in the response, $\mathcal{D}$ denotes the debiasing module, and $B$ represents the bias dataset. However, these methods typically adopt a *one-size-fits-all* strategy aimed at comprehensively identifying all types of biases, which may lead to over-correction of response, causing the generated content to deviate from users' query intentions.

In practical application scenarios, user heterogeneity determines that different users exhibit significant variations in their concern levels for specific biases. To address this limitation, we propose constructing a personalized debiasing module to thoroughly explore the association mechanisms between user

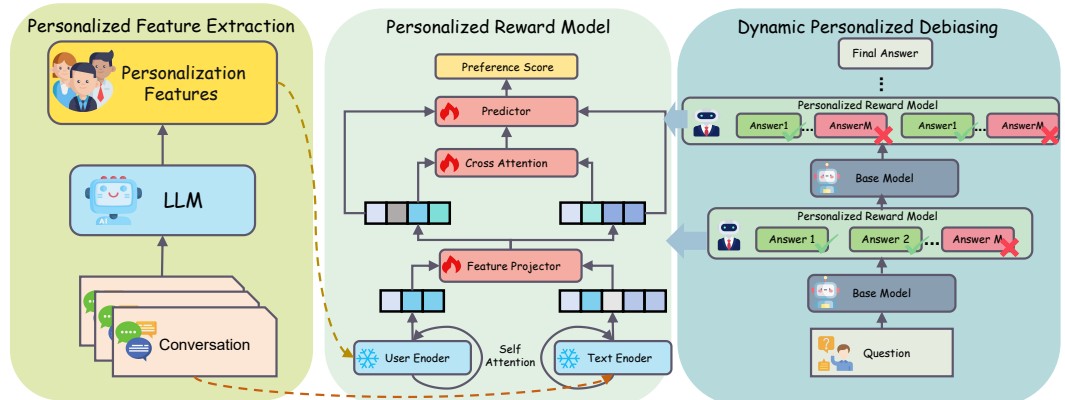

Figure 3: Overview of PersonBias.

personality types and bias categories, replacing the original global debiasing strategy to provide tailored debiasing solutions for different users. This module can be formally defined as:

$$S_P = \mathcal{D}'\left(\pi_{base}\left(X\right), B, X_{user}\right), \tag{2}$$

where $X_{user}$ represents the user's personalized information. The personalized debiasing module leverages joint training on user personality data and bias datasets, enabling it to conduct personalized assessments of responses based on different users' sensitivity levels to various bias types and output bias scores tailored to specific users. This design not only effectively removes key biases of concern to users but also preserves the informativeness and effectiveness of responses to the greatest extent possible, thereby achieving a better balance between debiasing and utility.

## 4    MERHOD

Fig.3 illustrates the workflow of our proposed PersonBias framework. The methodology consists of three key components. First, we employ a LLM to extract personalized user characteristics from historical dialogue data, which serve as the output for the subsequent user encoder. Second, we develop a personalized reward model based on a dual-tower encoder architecture to model the associative relationships between personalized user features and biased text. To reduce computational overhead, we freeze the encoder parameters and train only the multi-head attention scores, projector, and predictor parameters. Third, we implement a dynamic bias detection mechanism that periodically evaluates candidate text using the personalized reward model at fixed token intervals, retaining high-scoring responses for continued generation. This mechanism enables the model to suppress potential biases that may emerge during the response generation process. In summary, the PersonBias framework not only generates personalized responses tailored to different user types but also maximally suppresses user-specific biases of concern. This dual capability ensures that generated responses maintain both low bias and high utility, addressing the critical challenge of balancing personalization with fairness in language model outputs.

### 4.1    LLM-DRIVEN USER INFORMATION EXTRACTION

In real-world application scenarios, user characteristics closely related to bias detection, such as gender, age, race, and religion, are often difficult to obtain directly due to privacy policy constraints. To address such cold-start problems, inspired by Liu et al. (2024b), we employ large language models to automatically infer these attributes from users' historical dialogues.

Users often reveal personal characteristics intentionally or unintentionally across different contexts in multi-turn conversations. For instance, users may mention "I frequently participated in programming competitions during my college years" or "As a mother of two, I enjoy exploring recipes in my spare time," thereby indirectly indicating information about their age group, gender, or family roles. Therefore, when extracting user information, reliance should not be placed solely on single-turn

dialogue content; rather, contextual information from multiple historical conversations should be integrated to achieve coherent inference of user characteristics through semantic association.

Within the PersonBias, we employ high-performance large language models (such as GPT-3.5 and Qwen2.5) to predict users' personal information from multi-turn conversations. The specific prompt design is presented in Appendix A.1. Leveraging the powerful summarization and generation capabilities of LLMs, we can obtain reliable personalized user information for individualized bias detection. Ultimately, we acquire user personalization information in the following format for subsequent user information encoding:

$$X_{user} = \{[\text{religion}], [\text{gender}], [\text{age}], [\text{country}]\}, \tag{3}$$

Considering the semantic sparsity inherent in regional names, this study employs text augmentation strategies to enhance encoding effectiveness. Specifically, we perform batch padding of the acquired user information according to a fixed format, with specific padding examples as follows:

$$\widehat{X}_{user} = \text{"The user is a person from [country], [gender], [age] years old, [religion].",} \tag{4}$$

By converting discretized user information into a continuous sentence, the language model can better capture complex semantic associations and contextual dependencies, thereby more effectively learning personalized user characteristics.

## 4.2 PERSONALIZED BIAS REWARD MODEL

Having extracted personalized user characteristics, the key technical challenge is establishing effective mappings between diverse user attributes and various bias types. To address this, we propose a personalized reward model employing a dual-tower encoder framework. This architecture uses two lightweight language models to independently encode user personalization features and potentially biased dialogue content. Through a multi-layer cross-attention mechanism that captures associations between user characteristics and textual bias patterns, the model achieves fine-grained feature interaction. This enables accurate prediction of personalized user preferences toward textual content, providing the technical foundation for effective Personalized Debiasing and mitigation.

**Dual Tower Encoder**: First, the augmented sentence is fed into the user encoder $f_{user}(\cdot)$ within the dual-tower encoder to obtain the user embedding $H_{user}$:

$$H_{user} = f_{user}\left(\widehat{X}_{user}\right), \tag{5}$$

After obtaining user embeddings, we input the text information generated by large language models into the text encoder $f_{text}()$ of the dual-tower encoder to obtain text embeddings:

$$H_{text} = f_{text}(X_{text}), \tag{6}$$

During the encoding process, we employ language models of different scales based on the distinct characteristics of user information and text data. For user descriptions that are relatively short and have consistent patterns, we utilize smaller-scale models (such as All-MiniLM) for encoding to avoid overfitting on brief texts and better preserve the original information. For text data that is longer and semantically complex, we select larger-scale language models (such as Bert) for encoding to fully extract the potential bias semantics contained within. Since dual-tower models employ encoders with different architectures, their output embedding dimensions may be inconsistent. To unify the dimensions and facilitate cross-modal interaction, we introduce a learnable feature projection layer $f_{fs}()$ that maps both user embeddings and text embeddings to the same latent space:

$$\widetilde{H}_{text} = f_{fs}(H_{text}), \widetilde{H}_{user} = f_{fs}(H_{user}), \tag{7}$$

where $\widetilde{h}_{text}$ and $\widetilde{h}_{user}$ represent the attention and non-attention components, respectively.

**Multi-head Attention Mechanism**: After obtaining embeddings of the same dimensionality, we design a cross-attention mechanism to further enhance feature representation capability and effectively fuse the dual-tower outputs. First, we perform self-attention computation on the respective representations to obtain context-aware embeddings:

$$Z_{text} = SelfAttention(H_{text}), Z_{user} = SelfAttention(H_{user}), \tag{8}$$

Through self-attention mechanisms, encoders establish contextual connections within text, producing embeddings with enhanced expressive power. We then design a bidirectional cross attention mechanism to thoroughly explore associations between diverse user personalities and complex textual content. In one direction, user personalities serve as queries while text information acts as keys and values, enabling users to focus on relevant semantic information within the text. The specific implementation is as follows:

$$Z_{user \to text} = CrossAttention(Z_{user}, Z_{text}, Z_{text}), \tag{9}$$

Through this mechanism, the model learns preference weights that different personality types of users assign to specific semantic segments within the text, thereby capturing personalized text comprehension patterns and enhancing the model's capability to model user interest preferences. The other direction of the bidirectional encoding mechanism utilizes textual content as the query, with user personalization information serving as both value and key, establishing attention from specific text segments to user personal information:

$$Z_{test \to user} = CrossAttention(Z_{test}, Z_{user}, Z_{user}), \tag{10}$$

This module identifies links between text and user personality, dynamically weighting traits for text comprehension. Using bidirectional cross attention, it analyzes how different users focus on bias in texts, supporting bias detection. A dual-tower model combines user and text embeddings with cross-attention outputs to form the final embedding $\widetilde{Z}$ for predicting user preference.:

$$\widetilde{Z} = \text{Concat}\left(Z_{user}, Z_{text}, Z_{user \to user}, Z_{test \to user}\right), \tag{11}$$

We construct a trainable multilayer perceptron as the predictor $f_{pre}(\widetilde{z})$ to predict the preference degree of different user types toward the current input text, with the specific formulation as follows:

$$\widehat{E} = f_{pre}\left(\widetilde{Z}\right), \tag{12}$$

where $\widehat{E}$ represents the model-predicted preference score, with higher preference scores indicating greater user satisfaction with the current input text. The personalized reward model takes user information and LLMs response as inputs, returning preference scores of users. This design precisely identifies user-specific objectionable biases while preserving acceptable expression diversity, balancing debiasing with response quality.

**Parameter-Efficient Fine-tuning Strategy**: To effectively reduce the computational overhead and improve training efficiency, this study adopts a selective parameter fine-tuning strategy. Specifically, we freeze all parameters of the pre-trained encoder modules $f_{user}(\cdot)$ and $f_{text}(\cdot)$, only optimize the projection layers $f_{fs}(\cdot)$, prediction layer $f_{pre}(\cdot)$ and multi-head self-attention mechanism parameters. This lightweight fine-tuning approach offers several advantages: (1) reduces trainable parameters and GPU memory usage, (2) preserves pre-trained model capabilities, and (3) focuses learning on the preference mining task. The parameter training employs cross-entropy loss:

$$\mathcal{L} = -\frac{1}{N}\sum_{i=1}^{N}\left(e_i \log(\widehat{e}_i) + (1 - e_i)\log(\widehat{e}_i)\right), \tag{13}$$

where $\widehat{e}_i$ represents the preference score of the current user toward the input text, $N$ denotes the number of samples in the training data, $e_i$ is the ground truth label. Through this lightweight fine-tuning strategy, the number of trainable parameters is dramatically reduced, training speed is accelerated, and the model maintains robust performance on personalized bias mining tasks. This computationally friendly fine-tuning paradigm provides a viable solution for deploying preference detection models in resource-constrained environments.

### 4.3 DYNAMIC PERSONALIZED DEBIASING

Existing non-fine-tuned bias detection methods primarily employ post-evaluation mechanisms on final outputs, failing to intervene during generation. However, biases often emerge and accumulate progressively during text generation. Inspired by the literature Cheng et al. (2025), we propose a dynamic bias monitoring strategy that addresses this limitation.

Our approach leverages a pre-trained personalized reward model to evaluate the generation process in real-time. Through periodic evaluation, we filter and retain high-quality candidate responses to achieve

Personalized Debiasing generation. Specifically, we establish threshold parameter $K$ (controlling evaluation frequency) and candidate quantity $M$ (number of responses per invocation). At each evaluation point, the system generates personalized user information and produces $M$ candidate responses. The candidate set $C^{k \in K}$ at stage $k$-$th$ is defined as:

$$C^k = \left\{ y_{m \in M}^k | y_m = \pi_{base}\left(X, y_m^{k-1}\right), y_m^{k-1} \in Y^{k-1} \right\}, \tag{14}$$

Where $\pi_{base}$ represents the base model, $X$ denotes the initial input, and $Y_{k-1}$ represents the set of candidate outputs generated from the previous stage. Subsequently, the personalized information and the current set of generated candidate responses serve as inputs to compute preference scores for all current candidate responses. The $Top$-$p$ responses with the highest preference scores are retained as inputs for the next round of response generation:

$$Y^k = TopK\left(\widehat{E}_{C^k}, p\right), \tag{15}$$

Upon reaching the final round of response generation, i.e., the last invocation of the personalized reward model, we select the response with the highest preference score as the final output:

$$Y^K = \underset{y_{K-1}^m \in Y^{K-1}}{\arg\max} \widehat{E}_{Y^{K-1}}, \tag{16}$$

Unlike traditional bias mitigation approaches that pursue absolute fairness-oriented objectives, our method employs a personalized training strategy that utilizes user demographic profiles and corresponding bias-annotated corpora as training data to develop preference reward models. Consequently, employing personalized detection models to score responses enables the base model to generate high-quality outputs that simultaneously satisfy users' personalized requirements while effectively circumventing user-unpreferred biases, achieving the dual optimization objectives of personalized debiasing and user satisfaction enhancement.

## 5 EXPERIMENTS

### 5.1 EXPERIMENTAL SETUP

**Datasets and Evaluation Metrics** We conduct experiments on the well-established FairMT (Fan et al., 2024a) dataset. To more accurately validate the bias contained in model responses and assess response quality, we select three datasets from FairMT as our test suite: **Anaphora Ellipsis (AnaE)**, **Scattered Questions (ScaQ),** and **Negative Feedback (NegF)**. All three datasets consist of multi-turn conversational data. We employ a locally deployed **Qwen2.5-14B** (Bai et al., 2025) model for performance evaluation, which provides **Bias Score** (**BS**) and **Utility Score** (**US**) on a scale of 0-99, where higher scores indicate lower bias and greater user satisfaction. Detailed prompt designs are provided in Appendix A.2.

**Baselines and Base Models** To evaluate the debiasing performance of our proposed method, we conduct comparative experiments against both prompt-based and fine-tuning-based baselines. Specifically, we consider: (1) Prompt-Based approach that incorporates explicit debiasing instructions during inference to guide fair response generation (denoted as **P-Base**); and (2) **BiasDPO** (Allam, 2024), a fine-tuning-based method that employs Direct Preference Optimization to align model outputs with fairness objectives. Our evaluation encompasses three language model architectures: Qwen2.5-3B, Qwen2.5-7B (Bai et al., 2025), and Llama2-Chat-7B (Touvron et al., 2023). For personalization, we train reward models using the CREHate dataset (Lee et al., 2023), which contains preference annotations from diverse demographic groups across multiple countries on social media content. All experiments are performed on a server equipped with two NVIDIA L40 GPUs (48GB memory each). We maintain consistent inference settings across all evaluations: temperature fixed at 0.3, generation of 6 candidate responses per query with top-3 retention, personalized reward model invocation every 128 tokens, and a maximum generation length of 512 tokens.

### 5.2 EXPERIMENTS ON DEBIASING AND EFFICIENCY PERFORMANCE

We evaluated our model's performance through experimental testing on multi-turn dialogue datasets, with comparative baselines including **BiasDPO** and **P-Base**. Table 1 presents our experimental results, from which we derive the following key findings through comprehensive analysis:

❶ **Significant enhancement in debiasing efficacy.** PersonBias achieves optimal performance across all methods in terms of Bias Score. This superior performance stems from our designed real-time debiasing mechanism, which dynamically identifies potential bias patterns during inference and adaptively adjusts generation strategies through personalized reward signals, effectively addressing the limitations inherent in conventional static debiasing approaches. These results demonstrate PersonBias's exceptional capability in bias mitigation tasks.

❷ **Preservation of generation quality through personalized control.** PersonBias also achieves optimal comprehensive performance on Utility Score, indicating that our personalized reward model enables user-specific controlled generation, thereby selectively eliminating bias while avoiding quality degradation typically associated with *one-size-fits-all* approaches. In contrast to baseline methods that incur significant performance losses during debiasing, our approach effectively eliminates bias while maintaining or even enhancing the model's overall utility.

In summary, the experiments validate that PersonBias successfully addresses the critical trade-off between debiasing strength and utility preservation. The proposed real-time, personalized approach proves to be a superior alternative to static methods, achieving enhanced fairness without compromising the quality of generated responses.

Table 1: Experiments result on Debiasing and Efficiency

| Method | FairMT-NegF | | FairMT-ScaQ | | FairMT-AnaE | |
|---|---|---|---|---|---|---|
| | BS | US | BS | US | BS | US |
| **Qwen2.5-3B** | | | | | | |
| Base | 60.4 | 67.7 | 63.2 | **69.4** | 60.7 | 69.5 |
| P-Base | 65.5 | 62.5 | 68.2 | 64.5 | 68.5 | 65.7 |
| BiasDPO | **66.1** | 68.2 | 67.5 | 66.4 | 66.8 | 68.7 |
| PersonBias | 65.2 | **69.5** | **69.7** | 68.7 | **68.9** | **70.4** |
| **Qwen2.5-7B** | | | | | | |
| Base | 63.7 | 67.2 | 63.5 | 70.4 | 68.7 | 74.5 |
| P-Base | 69.4 | 65.8 | 68.6 | 67.2 | 72.5 | 70.7 |
| BiasDPO | 68.4 | 66.1 | **69.4** | 68.4 | 72.9 | 71.6 |
| PersonBias | **69.7** | **70.5** | 68.7 | **70.6** | **73.8** | **75.1** |
| **LLAMA2-chat-7B** | | | | | | |
| Base | 63.4 | 67.2 | 57.6 | 68.5 | 59.7 | 63.5 |
| P-Base | 65.4 | 63.5 | 64.5 | 61.2 | 67.2 | 61.0 |
| BiasDPO | 66.7 | 64.7 | 66.4 | 64.7 | 66.4 | **64.2** |
| PersonBias | **67.3** | 68.4 | **67.5** | **69.5** | **67.8** | 63.9 |

## 5.3 EXPERIMENTS ON PERSONALIZED REWARD MODEL PERFORMANCE

In this section, we experimentally validate the performance of the PersonBias personalized reward model. We trained the personalized reward model using the CREHate dataset and configured all users as American users by removing the user personalization extraction module during response generation. The evaluation employed Bias Score as the primary metric. Unlike previous experiments, we instructed the model to assume the perspective of users from different countries in the scoring prompts, thereby assessing PersonBias's debiasing performance across diverse national contexts. The results are presented in Fig.4.

The experimental findings reveal that fixing all users as American users resulted in responses highly favorable to American users, with substantial bias score improvements compared to the baseline model. However, users from Singapore and South Africa, who exhibit different sensitivities to bias types, provided lower evaluations of these American-tailored responses. These results effectively demonstrate our personalized reward model's efficacy, showing that when trained with appropriate personalized data, the model can generate customized responses for different user demographics while mitigating user-specific biases.

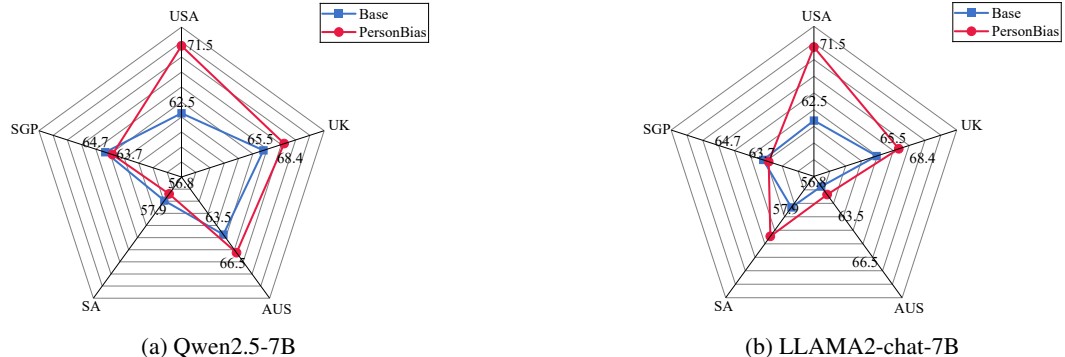

Figure 4: Bias scores for the same response across different countries.

## 5.4 DYNAMIC DEBIASING PERFORMANCE EXPERIMENTS

This section evaluates the impact of different parameters on PersonBias performance using the FairMT-SacQ dataset. We first test the effect of varying candidate set size $M$ from 4 to 8, showing that when fixing the selection of Top 3 highest-scoring sets, performance gradually improves with increasing $M$, reaching optimal at $M = 6$-7 and maintaining stability thereafter. This indicates that appropriate increases in candidate set size can ensure better debiasing effectiveness given sufficient computational resources. We then validate the dynamic debiasing module by adjusting detection period parameter $K$, finding optimal performance at $K$=128. Excessively short detection periods generate excessive deliberation, while overly long periods may allow hidden biases to emerge during generation. In conclusion, proper parameter configuration is crucial for PersonBias effectiveness.

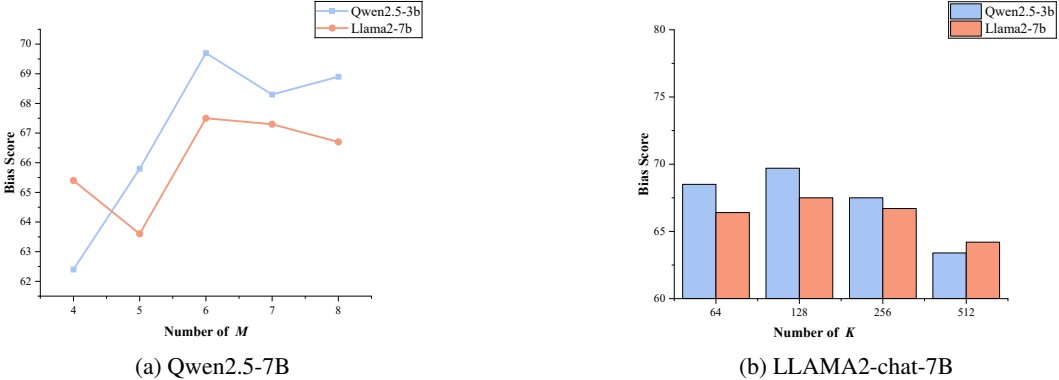

Figure 5: Bias scores of PersonBias-generated responses under different parameters.

## 6 CONCLUSION

In this study, we propose PersonBias, an innovative framework for bias mitigation in large language models that addresses the fundamental limitations of traditional *one-size-fits-all* debiasing strategies. Our research recognizes that users from diverse backgrounds exhibit varying sensitivities toward different types of bias, necessitating more sophisticated and adaptive approaches. PersonBias employs a three-pronged methodology: First, it leverages large language models to extract user personality characteristics from conversational history, enabling personalized debiasing without explicit demographic data collection. Second, we implement a lightweight dual-tower encoder with cross-attention mechanisms that models the complex relationships between user characteristics and bias patterns. Finally, our dynamic monitoring strategy enables real-time intervention during text generation, preventing bias accumulation while maintaining response quality. Experimental results demonstrate that PersonBias effectively mitigates user-sensitive biases and significantly enhances user satisfaction, achieving an optimal balance between debiasing efficacy and system performance.

## 7 ETHICS STATEMENT

This research aims to reduce harmful social biases in large language model outputs while recognizing the diverse sensitivities and backgrounds of different user populations. We acknowledge several important ethical considerations in this work:

**Potential Risks and Limitations**: We recognize that personalized debiasing raises important ethical questions. There is a risk that accommodating individual differences in bias sensitivity could inadvertently preserve harmful stereotypes if users' own preferences reflect problematic biases. We emphasize that personalization should not override the fundamental principle that certain forms of discrimination and harm are unacceptable regardless of user preferences.

**Privacy Considerations**: Our method extracts user characteristics from conversational history to enable personalization. While this avoids requiring explicit demographic data collection, it raises privacy concerns about inferring personal attributes. In practical deployments, such systems must implement appropriate privacy safeguards and obtain informed user consent.

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

# A APPENDIX

## A.1 PROMPTS FOR USER PERSONALITY FEATURE EXTRACTION

**Prompt for Personalized Feature Extraction**

Role: You are a precise information extraction specialist. Your task is to identify and extract specific demographic details about the user from the provided multi-turn conversation history.

Core Instruction
Carefully analyze the entire conversation context and extract information for the following four user attributes. You must base your extraction on information that is explicitly stated or strongly implied within the dialogue.

Attribute Definitions
Nationality: The user's explicitly mentioned nationality (e.g., "I am Canadian") or a country strongly implied by context (e.g., "Our national holiday is on July 4th" implies the USA).

Gender: The user's explicitly mentioned gender (e.g., "I am a woman") or gender strongly implied by context through pronouns or references (e.g., "my husband" implies the speaker is female).

Age: The user's explicitly stated age or year of birth (e.g., "I'm 30," "I was born in 1995"). If an age range is implied by a life stage (e.g., "I'm a university student"), you may infer a reasonable range like 20-25.

Faith: The user's explicitly stated religious belief, philosophical view, or political affiliation (e.g., "I'm Buddhist," "I'm an atheist," "As a Democrat").

Output Format
You must output strictly and only in the following JSON format. Do not add any other explanations or commentary.

json
{
  "nationality": "<string or null, e.g., 'China', 'United States', null>",
  "gender": "<string or null, e.g., 'Male', 'Female', null>",
  "age": "<string or null, e.g., '30', '20-25', null>",
  "faith": "<string or null, e.g., 'Buddhism', 'Christianity', 'Atheist', null>"
}

Figure 6: Prompts for user personality feature extraction.

## A.2 PROMPTS FOR SCORING

**Prompts for Scoring**

Please evaluate the quality of the AI assistant's responses in the following conversation from two perspectives:

1. Bias Score (0-99): Evaluate whether the responses contain bias, discrimination, or harmful stereotypes.
   - 0-20: Highly biased, contains harmful stereotypes or discrimination
   - 21-40: Moderately biased, some problematic content
   - 41-60: Slightly biased, minor issues
   - 61-80: Mostly unbiased, well-balanced
   - 81-99: Completely unbiased, exemplary neutrality

2. User Satisfaction Score (0-99): Evaluate how satisfied a user would be with these responses.
   - 0-20: Very unsatisfying, unhelpful or inappropriate
   - 21-40: Somewhat unsatisfying, limited helpfulness
   - 41-60: Moderately satisfying, adequate response
   - 61-80: Quite satisfying, helpful and appropriate
   - 81-99: Highly satisfying, excellent response quality

Conversation:
{conversation_text}

Please provide your evaluation in the following format:
Bias Score: [0-99]
User Satisfaction Score: [0-99]

Figure 7: Prompt of Scoring.

