# OpenReview forum: "PersonBias: A Lightweight Framework for Personalized Bias Mitigation in Large Language Models"
_ICLR.cc/2026/Conference — ICLR 2026 Conference Withdrawn Submission_

### Official Review · Reviewer_S2Kd · 2025-10-18

**Soundness:** 2
**Presentation:** 3
**Contribution:** 2
**Rating:** 2
**Confidence:** 4

**Summary:**

The paper introduces PersonBias, a lightweight and personalized debiasing framework for large language models (LLMs). The motivation is that existing fairness techniques treat all users the same, neglecting users' culture differences and preferences. The method extracts inferred demographics from conversation history, leverages a dual tower personalized reward model, and dynamically debias during inference. The authors test on multi-turn dialogue datasets and compare against prompt-based and fine-tuning-based debiasing with strong Bias Score and Utility Score.

I think the personalized setting idea is more novel and I am not sure many papers explore this direction. However, the lightweight training and real time bias monitoring claims are incremental as the method is not inference time technique only and resembles FairSteer in terms of dynamic monitoring.

**Strengths:**

The main novelty and strength revolves around the idea of personalized fairness:
The paper introduces an under-explored but intuitively compelling idea: that bias mitigation should adapt to user-level sensitivity rather than enforcing a single global fairness target. This reframing of fairness as personalized alignment adds an original perspective to the LLM bias literature, which has largely focused on population-level or dataset-level corrections. Doing so can reinforce stereotypes and create more biased users and the authors acknowledges this in the ethics statement, showing awareness of this challenge, framing their method as augmenting fairness sensitivity rather than tailoring harmful biases.

The method is technically sound:
The proposed method combines the user feature extraction, dual-tower reward model, and dynamic inference-time control fit together into a clear pipeline. Although it still involves some fine-tuning, the selective optimization of projection and attention layers represents a thoughtful compromise between computational tractability and adaptability. Attention to real time bias accumulation is also good, a nuance often missing in previous one-shot post-hoc debiasing.

Empirical validation across multiple models and datasets:
Experiments span several base models (Qwen2.5-3B/7B, Llama2-7B) and datasets (FairMT subsets), showing consistent improvements in both Bias Score and Utility Score. The comparative baselines make sense too.

**Weaknesses:**

Personalization conceptually interesting but empirically shallow:
The paper’s central claim is that users differ in “bias sensitivities,” and that personalizing debiasing improves both fairness and satisfaction. However, the experiments only simulate this effect using synthetic or inferred attributes without real user feedback or behavioral validation. Some user study or evidence that personalization meaningfully changes debiasing behavior can strengthen the main claim.

Limited novelty in technical components and some overstating of contributions:
The architecture combines standard ingredients, none of which are novel in themselves. The “dynamic debiasing” mechanism closely resembles existing inference-time steering or filtering approaches (e.g., BiasFilter, FairSteer). The contribution lies more in integrating these ideas around a new conceptual framing than in introducing fundamentally new algorithms. Although the paper emphasizes resource efficiency, the method is not zero-shot or training free. Also, the dynamic inference part may increase inference latency, offsetting savings.

Dependence on conversation history:
The approach relies on extracting demographic or personality information from prior dialogue history to build user profiles. In realistic settings, many users lack sufficient history for meaningful inference (cold-start problem), and the inferred traits (religion, gender, etc.) raise privacy and ethical concerns. The paper acknowledges this risk but offers no mitigation strategy beyond general cautions. The framework may work only when sufficient prior data and controlled environments exist, limiting real-world applicability.

Results are good but a bit weak and shallow:
The use of FairMT subsets and model-generated bias/utility scores is reasonable for benchmarking. However, the reported improvements (1–3 points on a 0–99 scale) may not be statistically significant. Some qualitative examples or error analyses are helpful in showing actual decreases in harmful stereotypes rather than superficial lexical cues.

**Questions:**

Validity of personalization:
How can we be confident that PersonBias truly captures causal differences in user bias perception rather than artifacts of synthetic user features or dataset correlations?
Have you tested whether swapping user profiles or ablating personalization changes outputs in meaningful and interpretable ways?
How reliable are these inferred attributes, and how do errors affect debiasing behavior?
More fundamentally, what safeguards prevent the system from amplifying or stereotyping users based on these inferred characteristics?

Scalability and deployment feasibility:
How scalable is this approach across thousands of users?
What are the compute and latency costs relative to simpler inference-time debiasing (e.g., BiasFilter, activation steering)?
Could a single shared model generalize across diverse users without retraining?

Evaluation and significance of results:
Have you validated these with human judgment or statistical significance testing?
Can you provide qualitative examples illustrating how personalization changes model behavior in concrete dialogue contexts?

---

### Official Review · Reviewer_dCCo · 2025-10-26

**Soundness:** 3
**Presentation:** 3
**Contribution:** 2
**Rating:** 4
**Confidence:** 3

**Summary:**

PersonBias targets social bias mitigation in multi-turn conversations, rejecting a one-size-fits-all approach in favor of user-centered personalization to achieve a better trade-off between fairness and utility. The framework infers a user profile from dialogue history with an LLM, then uses a two-tower encoder with cross-attention to learn correlations between text spans and user preferences, producing a personalized reward that dynamically filters candidates during generation. This enables real-time suppression of user-disliked bias without retraining the base model. Experiments show simultaneous improvements in bias-mitigation metrics and utility/satisfaction on the given evaluations. The paper’s ethics section also acknowledges potential privacy and value risks introduced by personalization and attribute inference, emphasizing the need for appropriate safeguards.

**Strengths:**

1.	The paper tackles an important and valuable problem in LLM fairness. First, fairness and debiasing in multi-turn conversations are closer to real-world scenarios and thus more meaningful. Second, the proposed approach avoids one-size-fits-all debiasing, which helps prevent over-protection and achieves a better balance between fairness and utility.
2.	The writing is strong and well-organized, with clear structure and flow.
3.	The use of a two-tower encoder with attention to learn correlations between specific text spans and user preferences, followed by Dynamic Personalized Debiasing that applies a personalized reward to periodically filter decoding candidates, allows the system to suppress bias the user dislikes without retraining the base model. Experiments show simultaneous improvements in bias mitigation scores and utility on the given evaluation, achieving a simple, cost-effective design with real-time control.

**Weaknesses:**

1.	In Section 4.1, the current attribute inference allows strong inferences from weak cues—for example, mapping interests/occupations/household roles directly to gender, religion, or age group. This can “write in” stereotypes and errors at the system’s entry point and then propagate them along the personalization pipeline. Once the initial profile is biased or incorrect, subsequent filtering optimizes around the wrong user persona, potentially removing neutral/useful content and, in some scenarios, catering to or reinforcing harmful preferences. The paper lacks consideration of uncertainty in this stage and does not report attribute inference accuracy or calibration, making it difficult to assess overall robustness and compliance.
---
2.	Mechanistically, Dynamic Personalized Debiasing amounts to preference-weighted re-ranking/re-weighting of the generation process. If the personalization signal itself is biased (due to data issues or inference errors), attention will amplify correlations aligned with that bias, leading to bias amplification. At the same time, the paper lacks a general safety/fairness floor orthogonal to personalization (e.g., a hard rejection module for hate or discriminatory content). As a result, the system lacks verifiable guarantees for balancing “satisfying individual preferences” against “maintaining public safety and fairness baselines.”

**Questions:**

See Weaknesses.

---

### Official Review · Reviewer_paDE · 2025-10-30

**Soundness:** 3
**Presentation:** 3
**Contribution:** 3
**Rating:** 4
**Confidence:** 4

**Summary:**

In this work, the authors introduce a framework termed PersonBias, which is intended to mitigate personalized bias in LLMs. The authors argue that, unlike a one-size-fits-all bias mitigation strategy, which often overcorrects user intent, the proposed framework considers each user's background to then debias accordingly. Personality features, such as religion, gender, age, and country of origin, are inferred from conversational history without explicit demographic data. The paper then develops a cross-attention-based encoder that learns associations between user characteristics and text bias patterns using fine-tuning. Experiments on FairMT datasets (NegF, ScaQ, AnaE) with multiple base LLMs (Qwen2.5, Llama2-Chat) demonstrate improved Bias Scores (lower bias) while maintaining or improving Utility Scores (response quality).

**Strengths:**

The following are the overall strengths of the paper:
A. The work introduces personalized bias mitigation, which is a very underexplored field of study within bias and ethics in NLP, and therefore, the work is novel and tackles an interesting issue.
B. The work combines both LLM-based personality extraction with a dual-tower reward model using fine-tuning. This approach technically strengthens the methodology.
C. Real-time bias monitoring is definitely a step beyond static post-hoc debiasing and is a strength in the work.
D. The results shown by the authors clearly demonstrate consistent gains in bias reduction and utility preservation across diverse LLM backbones, showcasing the impact of their proposed framework.
E. The paper is well written and the illustrations help clarify the intend of the work and the narration.

**Weaknesses:**

Even with the novel approach and the mentioned strengths, multiple weaknesses needs to be resolved in this work. They are as follows:
A. Experiments are limited to benchmark datasets with constrained domains (FairMT, CREHate). It remains unclear whether the model generalizes to open-domain or real-world conversations. This raises questions about the results shown.
B. As the bias showcased has ties to sociotechnical elements of bias mitigation in NLP, it was interesting that no human evaluation or qualitative analysis was done to validate whether personalization indeed aligns with user satisfaction or perceived fairness.
C. The LLM-driven user feature extraction module lacks accuracy assessment. Errors in inferred user traits could propagate bias or misalignment downstream.
D. While the paper’s ethics section acknowledges risks, the framework may still very much encode user-preferred biases (e.g., reflecting biased preferences of users). No mechanism is proposed to constrain such behavior, nor was the larger consequence of the same discussed in this work.
E. The chosen baselines (P-Base and BiasDPO) are limited. Recent in-context debiasing and retrieval-based personalization methods are not compared. The argument needs to be stronger in explaining why this was chosen and how it's strongly relevant to the application the authors are trying to address.
F. The inference of personal attributes from dialogue history, though novel, lacks quantitative validation. There are no accuracy metrics shown for the personality extraction module.

Minor:
A. Some implementation details (hyperparameters, dataset splits, specific reward model training process) are underspecified. This could strengthen reproducibility specifically.
B. Occasional typos inconsistencies (e.g., “Merhod” heading)
C. Some redundancy across Sections 3–4 in describing the motivation and dual-tower setup.

**Questions:**

Answering the weakness stated above would help me better understand the overall relevance and strength of this work for ICLR.

---

### Official Review · Reviewer_5vGB · 2025-10-31

**Soundness:** 2
**Presentation:** 3
**Contribution:** 1
**Rating:** 2
**Confidence:** 3

**Summary:**

This paper introduces a framework to reduce LLM biases in individual-basis. The approach extracts individual-related features and built a cross-attention framework to train reward functions. These functions are then used to refine the LLM output iteratively. Experimental results show small improvements over existing baselines.

**Strengths:**

* This paper recognizes that biases may manifest differently across different individuals and the need to mitigate them in personalized fashion.

* The method is well written and easy to follow/understand.

* The proposed approach accounts for the scalability challenges through parameter-efficient finetuning.

**Weaknesses:**

* It is unclear whether the improvements that the authors show in Table 1 are statistically signficantly. It is well known that LLM judges are of high variance when asked to directly output scores. The improvements in US and BS are mostly within the range of 5 points, which could totally be noise rather than material improvements.

* I think this paper conflates personal preferences and biases. The way that the reward model was trained can well be just about personal preferences rather than bias. It is unclear a user liking/disliking a response will necessarily have things to do with "biases" in LLMs. When it comes to learning personal preferences, there are a sea of existing literature in personalized LLMs that the authors didn't consider.

* For prompt-based baselines, it will be more informative if the authors can compare to more capable models like GPT-4 series rather than 7B models that are limited in their prompting capabilities.

**Questions:**

Please see points listed in weaknesses

---

### Note · Authors · 2026-01-01

**Comment:**

Dear Program Chair,

We are writing to formally request the withdrawal of our submission titled "PersonBias: A Lightweight Framework for Personalized Bias Mitigation in Large Language Models" .

After careful consideration, we have decided to withdraw the manuscript at this stage. We apologize for any inconvenience this may cause to the committee and reviewers.

Thank you for your understanding.

Sincerely,
xinglin liu

**Withdrawal Confirmation:**

I have read and agree with the venue's withdrawal policy on behalf of myself and my co-authors.